# Survival of hibernating little brown bats that are unaffected by white-nose syndrome: Using thermal cameras to understand arousal behavior

**Haley J. Gmutza**[1,2]*, **Rodney W. Foster**[1], **Jonathan M. Gmutza**[3], **Gerald G. Carter**[2,4], **Allen Kurta**[1]

1 Department of Biology, Eastern Michigan University, Ypsilanti, Michigan, United States of America,
2 Department of Evolution, Ecology, and Organismal Biology, The Ohio State University, Columbus, Ohio, United States of America, 3 eScape Technology LLC, Sterling Heights, Michigan, United States of America,
4 Smithsonian Tropical Research Institute, Balboa Ancón, Panama

* Gmutza.2@osu.edu

**Data Availability Statement:** All data and R code to repeat the analysis is publicly available on Figshare: https://figshare.com/projects/Synchronized_arousals_help_explain_resistance_

## Abstract

White-nose syndrome is a fungal disease that has decimated hibernating bats from multiple North American species. In 2014, the invasive fungus arrived at a hibernaculum of little brown bats (*Myotis lucifugus*) inside the spillway of Tippy Dam, located near Wellston, Michigan, USA, yet surprisingly, this population has not experienced the declines seen elsewhere. Unlike a typical subterranean hibernaculum, light enters the spillway through small ventilation holes. We hypothesized that this light causes the hibernating bats to maintain a circadian rhythm, thereby saving energy via social thermoregulation during synchronous arousals. To test this idea, we used high-resolution thermal cameras to monitor arousals from October 2019 to April 2020. We found that arousals followed a circadian rhythm, peaking after sunset, and that most observed arousals (>68%) occurred within a cluster of bats allowing for social thermoregulation. These findings are consistent with the hypothesis that light-induced synchronized arousals contribute to the unprecedented absence of mass mortality from white-nose syndrome in this large population. Using light to maintain a circadian rhythm in bats should be tested as a potential tool for mitigating mortality from white-nose syndrome. More generally, studying populations that have been largely unaffected by white-nose syndrome may provide insight into mitigation strategies for protecting the remaining populations.

## Introduction

White-nose syndrome is a deadly disease that afflicts bats that hibernate in subterranean sites in the United States and Canada and typically results in population declines of ~90% within 1–3 years of the first infection [1]. This disease is caused by the invasive fungal pathogen *Pseudogymnoascus destructans* (*Pd*), which grows on the exposed skin of hibernating bats, often

to_white-nose_syndrome_in_a_population_of_hibernating_little_brown_bats/175716.

**Funding:** Funding was provided by a grant from the U.S. Fish and Wildlife Service (F20AP00002) to AK and a contract with the East Lansing Field Office of the U.S. Fish and Wildlife Service (140F0319P0104) (https://www.fws.gov/) to AK and D.M. Reeder. The funders had no role in study design, data collection and analysis, decision to publish, or preparation of the manuscript.

**Competing interests:** The authors have declared that no competing interests exist.

resulting in a characteristic fuzzy white muzzle, which gives the disease its name [2, 3]. Mortality results from infected bats waking from torpor about twice as often as normal, which depletes fat stores twice as fast, resulting in starvation by early-to-mid winter [2]. Since its introduction, *Pd* has spread across much of North America and caused massive mortality (e.g., over 6 million bats in the United States between 2006 and 2012 [4–7]). For example, little brown bats (*Myotis lucifugus*) were once the most common bat in eastern North America, but they now face local extinction in some places, are listed as endangered in Canada, and are under consideration for endangered species status in the United States [7, 8]. As white-nose syndrome continues to devastate populations of multiple bat species, there is an urgent need for methods to combat the disease and to understand how some bats survive it.

An exception to this pattern of decline is Tippy Dam, a hydroelectric facility near Wellston, Michigan, USA (44.26°N, 85.94°W), where bats overwinter in the hollow, concrete spillway [9]. From 2012–2023, Tippy Dam has maintained a stable population of 20,000–25,000 hibernating little brown bats, despite presence of *Pd* on the walls and on the bats since 2015 and probably 2014 [10] (Fig 1). Massive mortality of *Pd*-infected little brown bats at Tippy Dam has not occurred, despite warm internal temperatures (>10°C) during autumn that correlate with high severity of infection at other sites [11]. Meanwhile, populations of hibernating bats in Michigan's abandoned mines have suffered the expected 90% mortality since local arrival of *Pd* in 2014 [12]. Understanding why the bats at Tippy Dam are not suffering mortality from *Pd* could provide valuable insights into mitigating the impacts of white-nose syndrome.

One hypothesis for the lack of mortality of bats at Tippy Dam involves the relationship between social thermoregulation, synchronized arousals, and circadian rhythms. During winter, most temperate bats hibernate—dramatically reducing their body temperature and metabolic rate to conserve energy [13]. Periodic arousals from hibernation are necessary for drinking, excreting waste, and other maintenance functions [14], but these arousals are energetically costly, accounting for up to 84% of energy spent during 6–8 months of hibernation [15]. Outside hibernation, many bats obtain substantial energetic benefits by clustering [16]. During hibernation, when body temperature typically approximates ambient temperature,

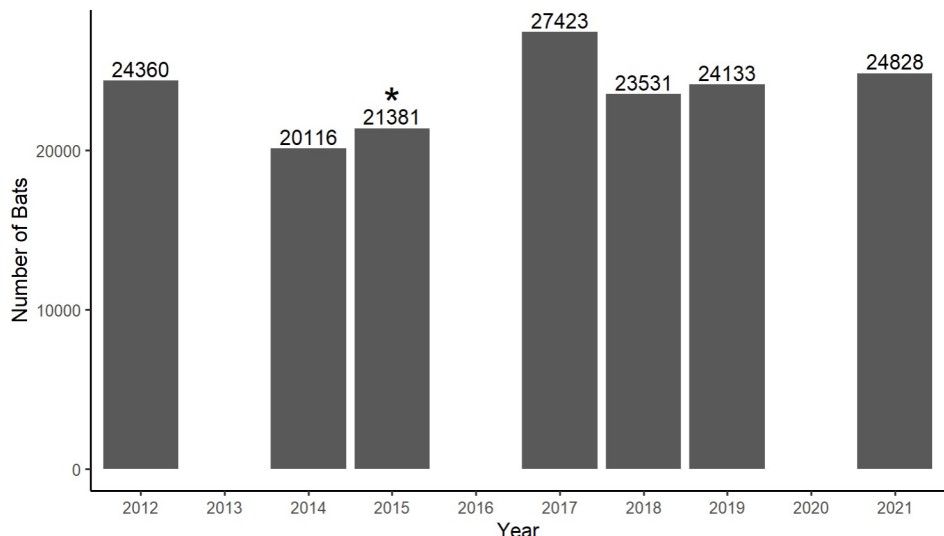

**Fig 1. Number of bats in Tippy Dam in 7 recent winters.** All counts were made in February of each year. Years with no bar are years in which counts were not performed. *Pseudogymnoascus destructans* was first confirmed in the spillway in 2015 (asterisk above bar) and was likely present in 2014 [10]. Historically, 97% of the population consisted of little brown bats [9].

most benefits garnered from clustering are realized during arousals, when animals increase their body temperatures and thus need to prevent loss of heat. Social thermoregulation during hibernation periods can therefore occur in the form of synchronized arousals among clustered conspecifics [17]. Waking as a group can be energetically beneficial to hibernators because arousing synchronously allows individuals to decrease their heat loss and benefit from passive rewarming [18, 19]. For example, clustered garden dormice (*Eliomys quercinus*) lost 2.5 times less body mass during a single arousal compared with solitary individuals [17].

Arousing with other bats is energetically beneficial but it requires a mechanism of synchrony. In mammals, many synchronous behaviors follow circadian rhythms, but these must be "entrained" by an external cue, called a zeitgeber, which is typically light. Without this entrainment, internal clocks stray from a 24-hour cycle [20], and most hibernating mammals, including bats, adopt a free-running rhythm, due to the constant darkness of their overwintering sites [21, 22]. However, thresholds for entrainment via light appear consistently low among bats, often near 0.02–0.0001 lux [23–25]. In fact, the lowest recorded threshold in a vertebrate is 0.00001 lux in Pallas's mastiff bat (*Molossus molossus*) [26]. Such low photic thresholds are presumably necessary for cave-dwelling bats to reset their clocks after they emerge from the absolute darkness of caves and experience only the dim light of dusk [26].

Little brown bats have been shown to lose their circadian rhythm when hibernating deep inside a cave [27], but Tippy Dam is unusual for a major hibernaculum in that bats are exposed daily to low levels of light entering through ventilation openings in the downstream wall of the spillway. If this dim light is a sufficient zeitgeber, then bats might stay on a circadian rhythm and arouse at night synchronously, in groups, thereby benefiting from social thermoregulation [16–19]. Synchronous arousals throughout winter could preserve energy (fat) stores, leading to a less-susceptible host.

Here, we investigated whether light entering a hibernaculum causes unusually synchronized arousals in the population of hibernating little brown bats at Tippy Dam. Using automated analyses of thermal video to determine the timing of winter arousals of the bats without disturbing them, we show that arousals are indeed synchronized, greatly increasing the potential for energy savings.

## Materials and methods

The spillway at Tippy Dam has multiple rooms, with ventilation holes allowing bats and light to enter, and is well protected from human disturbance [28]. We recorded bat arousals using thermal-imaging surveillance cameras (Model Q1942-E, Axis Communications, Lund, Sweden) that continuously monitored bats, in two rooms, over 174 days, from 23 October 2019 to 15 April 2020. Two light loggers (OMYL-M61, Omega Engineering, Norwalk, Connecticut) placed in the spillway in October 2019 failed to respond to the low levels of light in the dam during that winter, but light intensity was measured on 25 February 2021, between 1200 and 1230 h EST, with a dark-sky meter (Sky Quality Meter, Unihedron, Grimsby, Ontario; S1 Fig). We found the same arousal behaviors and results in both rooms, so we combined the data [29].

To automate counts of awake bats, we used a custom-made open-source program called MediaLab [30] to measure the location and area of warm bats in a video frame every 300 seconds. In thermal video, warm bats appear brighter than the colder backdrop of hibernating bats and the concrete wall, creating a measurable area of contrast. This contrast was used to detect the outlines of groups of awake bats, with the local time (EST) and date, area within the contour, and location of each contour within the frame automatically measured and recorded. This method easily discriminated among awake bats, the wall, and warm urine on the wall.

"Awake bats" were defined as areas of contrast with pixels that surpassed a specific brightness threshold. We estimated the minimum number of bats in a contoured warm spot by dividing the contour area by the average area of a single bat (based on a sample of 30 bats), and the brightness threshold for awake bats was determined similarly. The contour area estimates are minimum values, because multiple bats could be partly on top of each other and appear as one individual. Bats gain thermoregulatory benefits in clusters as small as 4 bats [31–33], and we arbitrarily defined an aroused "cluster" as 5 or more individuals that were normothermic simultaneously and in contact. Clusters containing fewer than 5 aroused bats were conservatively labeled as "solitary." See supplement for further description of the study site and video analysis.

If the light levels within the dam were enough to entrain the circadian rhythms of the bats, their arousals should consistently peak during night-time hours. To test this prediction, we used a permutation test to assess whether arousal frequency across the 24 hours of the day varied more than expected from randomly occurring arousals. Specifically, we first calculated the coefficient of variation (CV, standard deviation/mean) of counts of awake bats across hours. We then compared the observed CV to the expected distribution of CVs from when the observations of awake bats were permuted to random hours within the day (5,000 permutations). We calculated the p-value as the proportion of expected values that were greater than or equal to the observed value.

Our goal was to test the null hypothesis that the hibernating bats aroused at random times throughout the day, rather than a circadian pattern. As we did not have a precise prediction about when the peaks should occur or what form they should take, we did not create a statistical model of the nonlinear cyclical pattern. Rather, we directly convey the within-day magnitude and between-day consistency of the pattern by visualizing the arousal times both within and across days. For comparison, we then repeated this visualization with data expected from the null hypothesis.

When estimating cluster sizes of awake bats, we avoided repeatedly counting the same animals in consecutive frames. To do this, we randomly sampled frames with the restriction that no two frames occurred within 4 h of each other, because 3.9 h was the maximum arousal duration for bats within the dam during winter 2019–2020, based on data from temperature-sensitive radio transmitters on 37 bats (B. A. Daly in litt.). Repeating this random sampling four times with nearly identical results confirmed that these estimates of cluster sizes were robust. To estimate 95% confidence intervals around the mean cluster sizes, we used bootstrapping (5,000 iterations, percentile method, boot R package) [34, 35].

To describe how the timing of arousals differed with season, we conducted analyses separately for three hibernation periods: early (7 November–5 December 2019), middle (15 January–12 February 2020), and late (15 March–12 April 2020; S2 Fig). The early period had the highest temperatures, which decreased with time (from 8 to 4°C); the middle period had the lowest and most stable temperatures (2–3°C), whereas the late period was characterized by increasing temperatures (3–7°C; S2 Fig). Each period was 28-days long and separated by ca. 1 month. No human disturbance occurred inside the spillway during these times. Work was carried out under federal endangered species permit TE809630-4 from the U.S. Fish and Wildlife Service.

## Results

Consistent with a circadian rhythm, counts of awake bats were highly nonrandom, persistently lower during the day and peaking at night (Figs 2 and 3). Counts of awake bats across hours were more concentrated within certain hours than expected from chance in all three periods

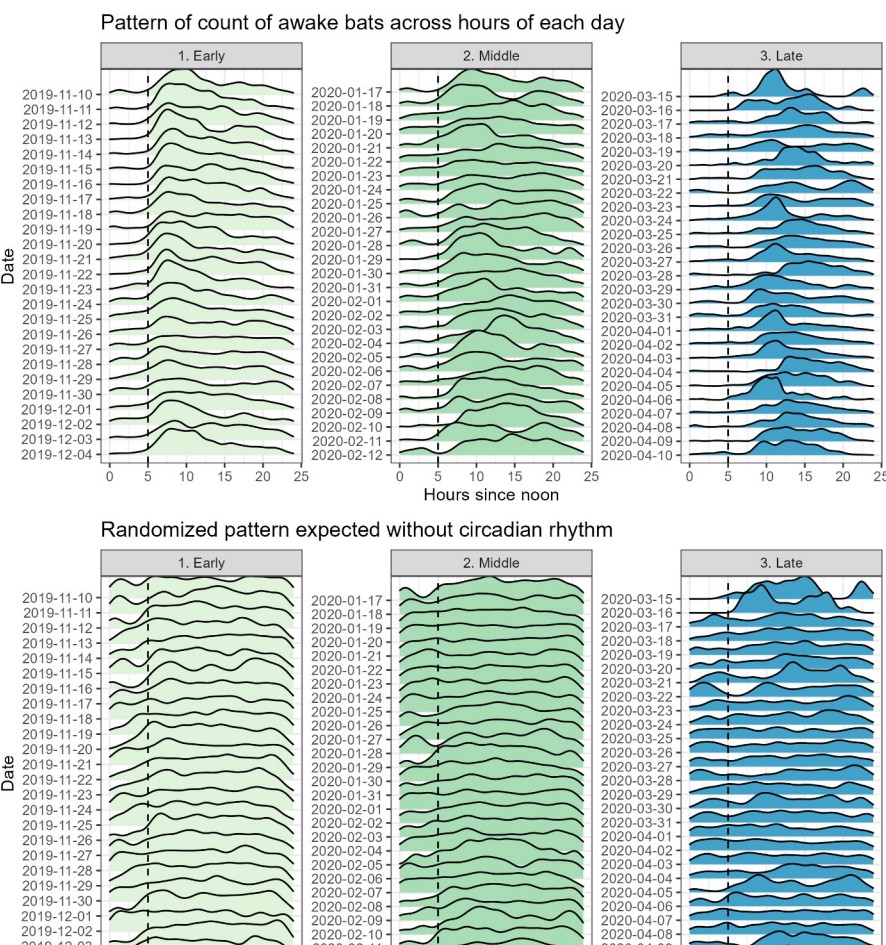

**Fig 2. Frequency of awake bats followed a circadian rhythm more than expected from random arousals.** The top three panels show the observed distribution of counts of awake bats over the course of one night for the early, middle, and late period. Each probability density curve shows the counts for one day. Bottom three panels show the same distributions expected from a random distribution of awake times for comparison. In the actual data (top panels), one can see striking patterns of (1) the circadian rhythm within each day and (2) the consistency and drift over time across days, as expected with the changing time of sunset and sunrise. Both these patterns are absent when arousals occur randomly within day (bottom panels). For reference, dashed vertical line is the earliest sunset at the site (5:05 PM EST December 5–16). Mean time of sunset for each period was 5:15 PM, 5:33 PM, and 7:08 PM, for the three periods, respectively.

(early: CV = 0.606, n = 12,650, p < 0.001; middle: CV = 0.468, n = 13,893, p < 0.001; late: CV = 0.727, n = 11,267, p < 0.001; Fig 2). The minimum estimate for the proportion of observed bats that aroused simultaneously within a cluster of 5 or more bats was 68.3% over the full hibernation season. Mean size of these clusters was 10.6 awake bats (95% CI = 10.55–10.65, n = 50,291), and maximum cluster size was 41 awake bats.

## Discussion

The little brown bats at Tippy Dam maintained a circadian rhythm leading to synchronized arousals, which should provide benefits from social thermoregulation, including reduced heat loss from insulation by adjacent conspecifics and heat gained from those bodies that are either warm or warming [19, 31–33]. The average cluster size of ca. 10 bats should be sufficient to

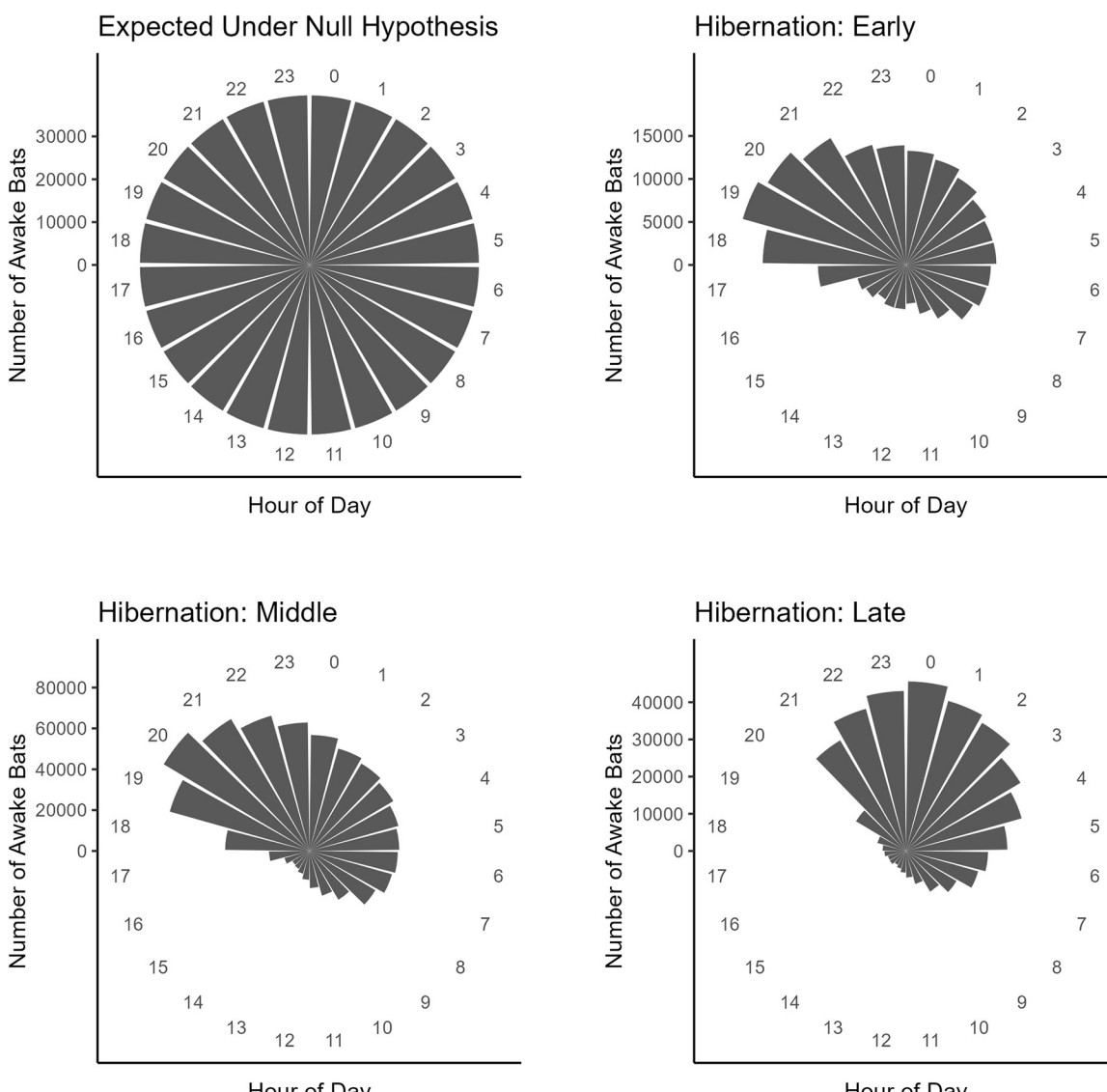

**Fig 3. Aggregated patterns show a circadian rhythm during each hibernation period.** Rose plots (circular histograms) show the aggregate number of awake bats during that hour over the whole period. Panels compare the expected distribution from the null hypothesis to the observed data for the three hibernation periods (early, middle, late).

reap energetic benefits from simultaneous arousals because the benefits of social thermoregulation can be gained in groups as small as 4 to 16 bats [36, 37]. Kurta (1985) used cooling curves of dead animals to show that a little brown bat in the center of a cluster of seven individuals had twice as much insulation available as did a solitary bat; doubling the available insulation in this way could theoretically reduce the energetic cost of an arousal by 50%, even if the adjacent animals were not normothermic [33]. Most observed arousals (>68%) occurred while an animal was in contact with at least four other aroused bats, suggesting that many synchronous arousals were due to arousal cascades in which bats awake sequentially due to disturbance from aroused conspecifics [38].

Our hypothesis that synchronized arousals explain lack of mortality is consistent with results from other studies. First, arousal activity by little brown bats at Tippy Dam was similar

to that of a population of Indiana bats (*M. sodalis*) that also showed both evidence of circadian rhythms and lack of mortality from white-nose syndrome [27]. Second, temperature-sensitive radiotransmitters revealed a lack of a circadian rhythm in uninfected little brown bats hibernating in Firecamp Cave in Manitoba [39]. The small entrance to that hibernaculum was covered by snow during winter, apparently blocking light and photic entrainment of arousal activity, showing that healthy little brown bats normally lack circadian rhythms in absence of light. Also, thermal cameras revealed that little brown bats exposed to *Pd* hibernating in complete darkness in a cave in Virginia lacked a circadian rhythm [27]. Future laboratory studies could help determine if arousing in small light-entrained groups contributes to enhanced survivorship and increased energetic benefits in bats impacted by white-nose syndrome. If synchronization of arousals is indeed a factor in the survival of these bats, then placing very dim light sources in hibernacula and operating these lights on a circadian pattern may be a potential management strategy.

Thermal imaging during this study provided unique insight into the activity of hibernating bats throughout an entire season. The data suggested that little brown bats at Tippy Dam, unlike those overwintering in caves, are using the available light to reset their internal clocks, which presumably facilitates synchronous arousals. Nevertheless, synchronous arousals may be only one component of the survival of this population [10]. For example, the interior of the spillway is too cold for rapid growth of *Pd* in mid-winter (2–3˚C) and too warm for fungal growth in summer, when ambient temperatures exceed 20˚C [40, 41] (B. Daly, in litt.; Fig S3). This may lead to the death of conidia, thus lowering the fungal load on the walls before the bats return in autumn [40]. In addition, the cutaneous microbiome of bats at Tippy Dam differs from that of animals hibernating in Kentucky or elsewhere in Michigan and may inhibit growth or reproduction of the fungus [42, 43]. More study is needed to determine how each of these factors might play a role in the persistence of this unique population. The impact of white-nose syndrome on multiple species of bats has been devastating [1, 5, 6, 12, 44], and what we learn from Tippy Dam may aid in combating progression of the disease, as it reaches caves and mines in western North America.

## Supporting information

**S1 Fig. Mean light values (lux) for each room in the spillway on 25 February 2021 were greater than the Erkert threshold of 0.0001 (black horizontal line).** Video was collected from rooms 8 and 9 (dark grey).
(TIF)

**S2 Fig. Graph of internal ambient temperature within Tippy Dam, winter 2019–2020.** The early (yellow), middle (brown), and late (pink) hibernation periods used in this study are shown via the colored rectangles. Data courtesy of B. A. Daly.
(TIF)

**S1 File.**
(DOCX)

## Acknowledgments

We thank B. A. Daly for the estimate of the maximum arousal duration from temperature-sensitive radio transmitters, as well as measurements of temperature and humidity within the dam, and for assisting with placement of the cameras and recorders. We also thank Consumers Energy, especially D. McIntosh and S. Whittaker, for allowing access to the spillway and providing logistical help.

## Author Contributions

**Conceptualization:** Allen Kurta.

**Data curation:** Haley J. Gmutza.

**Formal analysis:** Haley J. Gmutza, Gerald G. Carter.

**Funding acquisition:** Allen Kurta.

**Investigation:** Haley J. Gmutza.

**Methodology:** Haley J. Gmutza, Rodney W. Foster, Jonathan M. Gmutza.

**Project administration:** Allen Kurta.

**Software:** Jonathan M. Gmutza.

**Supervision:** Allen Kurta.

**Writing – original draft:** Haley J. Gmutza.

**Writing – review & editing:** Gerald G. Carter, Allen Kurta.

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
