## [Decision Letter · Decision Letter 0]

20 Oct 2023

PONE-D-23-28449Can synchronized arousals help explain why a population of hibernating little brown bats is unaffected by white-nose syndrome?PLOS ONE

Dear Dr. Gmutza,

Thank you for submitting your manuscript to PLOS ONE. After careful consideration, we feel that it has merit but does not fully meet PLOS ONE’s publication criteria as it currently stands. Therefore, we invite you to submit a revised version of the manuscript that addresses the points raised during the review process.

We look forward to receiving your revised manuscript.

Kind regards,

Karen Root, Ph.D.

Academic Editor

PLOS ONE

Journal Requirements:

3. We notice that your supplementary figures are uploaded with the file type 'Figure'. Please amend the file type to 'Supporting Information'. Please ensure that each Supporting Information file has a legend listed in the manuscript after the references list.

**Additional Editor Comments:**

This paper evaluates the effects of light on hibernating bats and their response to white-nose syndrome. It will make an important contribution to our understanding of how to manage habitat for vulnerable species, but the reviewers and I believe it needs some work in clarifying the approaches used, especially some of the underlying assumptions (e.g., energetic savings); improving the presentation of the results (e.g., Figure 2 could be simplified); and more emphasis on the strength of the paper (e.g., linking light to circadian rhythms observed and the resulting energy savings).  I agree with both reviewers that additional details are needed in the manuscript to better understand what can be inferred from these results and limit speculation on causation.  As Reviewer 2 suggests, while WNS is an important aspect to consider, the results are not sufficient to demonstrate a clear link between increased survival and the observed synchronized arousals.  You may need to add some additional caveats in the discussion to highlight the limits of the study (e.g., single study site with no control).  Both reviewers (and see the section below) have provided detailed suggestions on where improvements should be made.  Figure S3 was particularly helpful and illustrative of the unique aspects of this study and I would move it into the main body of the paper provide more description and discussion of them.  It would be helpful to place these results in the context of WNS mortality.  For example, if mortality occurs with WNS because infected bats are waking up twice as often as normal, how do the number of arousal events you measure compare?  Reviewer 1 also suggests an alternative way to evaluate the influence of winter periods, Julian day and hour is using GLMM, which could bolster your analysis and provide guidance as to critical features to further explore in future research.  A strength of the study is the exploration of circadian rhythms and synchronized arousals, but more is need to tie this directly to an increased survival with WNS.  The discussion should also be revised to better emphasize the contribution of this study and both reviewers provide guidance on what to emphasize. This paper has a lot of potential but needs some revision before suitable for publication.

Reviewers' comments:

Reviewer's Responses to Questions

**Comments to the Author**

1. Is the manuscript technically sound, and do the data support the conclusions?

Reviewer #1: Yes

Reviewer #2: No

2. Has the statistical analysis been performed appropriately and rigorously? 

Reviewer #1: Yes

Reviewer #2: Yes

3. Have the authors made all data underlying the findings in their manuscript fully available?

Reviewer #1: Yes

Reviewer #2: Yes

4. Is the manuscript presented in an intelligible fashion and written in standard English?

Reviewer #1: Yes

Reviewer #2: Yes

5. Review Comments to the Author

Reviewer #1: 136 - the p-value is not the frequency of observed vs expected but the probability that one is greater or lesser than the other. reword

138 - Fig 2 is a result, so why so early in the manuscript

153 - some analyses? Reword as the way it is written makes it sound like you were arbitrary, which you weren't

173 - no idea of the energetic savings ever given, but alluded to alot. Can you provide something to this? In theory and in reality this is probably true, but....

177 and 183 - unless I missed it, nothing in these papers (4 & &) about arousal synchrony or Indiana bats doing this

199 - good, so probably a Pd loading or lack thereof interaction. Perhaps a point to bring up in the introduction as well

Figure 2 - I get it but probably somehow combining for a mean observed line for early mid and late and a mean random line would be easier to read and get the point across. Right now you have to scan up or down, find a single date, look for a peak with the observed, then scan down to the random and find no peak. Got to be a better way. Also I wonder if you could do a GLMM or something on area or amount of warm pixels of observed versus random whereby you could better test winter period, julian day and hour among themselves to see how the observed patterns differ as time and photoperiod goes on? Implications there for hibernation length.

Reviewer #2: This is an interesting manuscript that offers valuable insights into specific light conditions that allow bats to synchronize arousals and conserve energy during rewarming in winter. I consider this to be the main strength of the study. I particularly appreciate the attempt made by the authors to link circadian synchronization and light conditions with white-nose syndrome. However, I believe that such link can not be directly proven with the current methodology used in the investigation. With this limitation, the causal factors enabling bats to be less susceptible to fungal infection remain unknown. I recommend that the authors focus the manuscript more on discussing arousal synchronization and its relation to energy savings in bats during the winter. This should be accompanied by a change in the manuscript’s title, avoiding the implication of a causal relationship in the white-nose syndrome. The tex body should be modified accordingly. Of course, the discussion of white-nose syndrome should be considered as one of the potential causes contributing to the synchronization, but not the sole consequence of the phenomenon. I would be happy to review the modified version of the manuscript.

6. PLOS authors have the option to publish the peer review history of their article (what does this mean?). If published, this will include your full peer review and any attached files.

Reviewer #1: No

Reviewer #2: **Yes: **Jorge Ayala-Berdon

---

## [Author Response · Author response to Decision Letter 0]

4 Dec 2023

Thank you for the comments! We are grateful that the manuscript was well received and we thank you and the reviewers for their insightful commentary. Our responses to the comments of the editor and reviewers are below.

Editor and reviewer comments are underlined.

Our responses are in bold.

Thank you,

Haley Gmutza, Rodney Foster, Jonathan Gmutza, Gerry Carter, and Allen Kurta

 

Editor

We agree that Figure S3 is more useful in the main text rather than the supplement. We have moved it into the manuscript as Figure 3, and have made a slight modification – instead of showing the aggregated data for the overall hibernation season in the top left corner, we have included a rose plot of the expected uniform distribution under the null hypothesis, to complement Figure 2, to highlight the fact that the length of the bars in each rose plot are scaled to the maximum value, and to show more starkly the circadian rhythm as a deviation from the data expected under the null model 

Revised figure and legend:

Figure 3. Rose plots showing aggregated arousals per hour for randomized data and the three hibernation periods (early, middle, late). Each bar represents aggregate number of aroused bats during that hour for the period.

Reviewer #1

136 - the p-value is not the frequency of observed vs expected but the probability that one is greater or lesser than the other. reword

Revised text: 

[144] “We calculated the p-value as the proportion of expected values that were greater than or equal to the observed value.” 

138 - Fig 2 is a result, so why so early in the manuscript

Fixed. We moved Figure 2 later to the results section of the manuscript.

153 - some analyses? Reword as the way it is written makes it sound like you were arbitrary, which you weren't

Fixed. We removed “some”.

Revised text: 

[162] “To describe how the timing of arousals differed with season, we conducted analyses separately for three hibernation periods: early (7 November–5 December 2019), middle (15 January–12 February 2020), and late (15 March–12 April 2020; Fig S2).”

173 - no idea of the energetic savings ever given, but alluded to alot. Can you provide something to this? In theory and in reality this is probably true, but....

We agree that we should elaborate more on energetic savings with regards to arousal. We added more information about energetic savings, clustering, and arousal to both the introduction and the discussion.

Added text [Introduction]: 

[76] “Periodic arousals from hibernation are necessary for drinking, excreting waste, and other maintenance functions [14], but these arousals are energetically costly, accounting for up to 84% of energy spent during 6–8 months of hibernation [15]. Outside hibernation, many bats obtain substantial energetic benefits by clustering [16]. During hibernation, when body temperature typically approximates ambient temperature, most benefits garnered from clustering are realized during arousals, when animals increase their body temperatures and thus need to prevent loss of heat. Social thermoregulation during hibernation periods can therefore occur in the form of synchronized arousals among clustered conspecifics [17]. Waking as a group can be energetically beneficial to hibernators because arousing synchronously allows individuals to decrease their heat loss and benefit from passive rewarming [18,19]. For example, clustered garden dormice (Eliomys quercinus) lost 2.5 times less body mass during a single arousal compared with solitary individuals [17].”

Added text [Discussion]: 

[202] “The average cluster size of ca. 10 bats should be sufficient to reap energetic benefits from simultaneous arousals because the benefits of social thermoregulation can be gained in groups as small as 4 to 16 bats [37,38]. Kurta (1985) used cooling curves of dead animals to show that a little brown bat in the center of a cluster of seven individuals had twice as much insulation available as did a solitary bat; doubling the available insulation in this way could theoretically reduce the energetic cost of an arousal by 50%, even if the adjacent animals were not normothermic [33]. Most observed arousals (>68%) occurred while an animal was in contact with at least four other aroused bats, suggesting that many synchronous arousals were due to arousal cascades in which bats awake sequentially due to disturbance from aroused conspecifics [39].”

177 and 183 - unless I missed it, nothing in these papers (4 & &) about arousal synchrony or Indiana bats doing this

We thank the reviewer for catching this mistake in referring to the wrong references. This text has been changed in the altered discussion (see above).

199 - good, so probably a Pd loading or lack thereof interaction. Perhaps a point to bring up in the introduction as well

We have added the following to the discussion, where temperatures inside the spillway are discussed in relation to fungal load:

[234] “This high summer temperature may lead to the death of conidia, thus lowering the fungal load on the walls before the bats return in autumn [41].”

Figure 2 - I get it but probably somehow combining for a mean observed line for early mid and late and a mean random line would be easier to read and get the point across. Right now you have to scan up or down, find a single date, look for a peak with the observed, then scan down to the random and find no peak. Got to be a better way. Also I wonder if you could do a GLMM or something on area or amount of warm pixels of observed versus random whereby you could better test winter period, julian day and hour among themselves to see how the observed patterns differ as time and photoperiod goes on? Implications there for hibernation length.

For clarity, we have included the following text in the paper:

[147] "Our goal was to test the null hypothesis that the hibernating bats aroused at random times throughout the day, rather than a circadian pattern. As we did not have a precise prediction about when the peaks should occur or what form they should take, we did not create a statistical model of the nonlinear cyclical pattern. Rather, we directly convey the within-day magnitude and between-day consistency of the pattern by visualizing the arousal times both within and across days. For comparison, we then repeating this visualization with data expected from the null hypothesis."

In our opinion, the most direct and powerful test of our null hypothesis is to simulate the outcomes of the null hypothesis directly using a custom null model which repeatedly permuted the bats' arousals to random times of the day, while controlling for all other structure in the data (number of arousals, number of bats per day, etc).

We also believe the visualization of the arousal times seen in Fig. 2 conveys both the circadian rhythm and the consistency across days better than either a single fitted curve or the parameter estimates of a multilevel model. By plotting density function within each day separately, we can see the sources of variation directly. In many cases, it is more insightful to fit a separate model to each random-effect group (in this case day) and then plot the effect across the groups to visualize the pattern.

For clarity, we added the following text to the plot legend:

"In the actual data (top panels), one can see striking patterns of (1) the circadian rhythm within each day and (2) the consistency and drift over time across days. Both these patterns are absent when arousals occur randomly within day (bottom panels)."

Reviewer #2

This is an interesting manuscript that offers valuable insights into specific light conditions that allow bats to synchronize arousals and conserve energy during rewarming in winter. I consider this to be the main strength of the study. I particularly appreciate the attempt made by the authors to link circadian synchronization and light conditions with white-nose syndrome. However, I believe that such link can not be directly proven with the current methodology used in the investigation. With this limitation, the causal factors enabling bats to be less susceptible to fungal infection remain unknown. I recommend that the authors focus the manuscript more on discussing arousal synchronization and its relation to energy savings in bats during the winter. This should be accompanied by a change in the manuscript’s title, avoiding the implication of a causal relationship in the white-nose syndrome. 

We thank the reviewer for their kind and constructive comments! We agree that the causal factors enabling bats to be less susceptible to fungal infection at this site remains unknown, and should have been more clear in the manuscript. We have revised the title and the text body to reflect this, and posit only that the link between circadian synchronization and white-nose syndrome survival is a testable hypothesis for the future.

Revised title: Survival of hibernating little brown bats that are unaffected by white-nose syndrome: using thermal cameras to understand arousal behavior 

The text body should be modified accordingly. Of course, the discussion of white-nose syndrome should be considered as one of the potential causes contributing to the synchronization, but not the sole consequence of the phenomenon. I would be happy to review the modified version of the manuscript.

We have modified the text body, particularly the discussion, to focus more on the synchronization element of the study and elaborate on energy savings gained from synchronization. The most major of these changes can be seen above. 

We do not mention white-nose syndrome as a possible cause of synchrony, because we are unaware of any evidence or rationale for why or how this would occur; however, we would be open to any evidence or argument for such a process. We thank the reviewer for being open to reviewing a revised version of the manuscript.

---

## [Decision Letter · Decision Letter 1]

15 Jan 2024

Survival of hibernating little brown bats that are unaffected by white-nose syndrome: using thermal cameras to understand arousal behavior

PONE-D-23-28449R1

Dear Dr. Gmutza,

We’re pleased to inform you that your manuscript has been judged scientifically suitable for publication and will be formally accepted for publication once it meets all outstanding technical requirements.

Kind regards,

Karen Root, Ph.D.

Academic Editor

PLOS ONE

Additional Editor Comments (optional):

I appreciate the authors’ thoroughness and thoughtfulness in addressing the numerous comments and suggestions by the reviewers.  With these revisions the paper is now suitable for publication and will make a notable contribution.

Reviewers' comments:

Reviewer's Responses to Questions

**Comments to the Author**

1. If the authors have adequately addressed your comments raised in a previous round of review and you feel that this manuscript is now acceptable for publication, you may indicate that here to bypass the “Comments to the Author” section, enter your conflict of interest statement in the “Confidential to Editor” section, and submit your "Accept" recommendation.

Reviewer #1: All comments have been addressed

2. Is the manuscript technically sound, and do the data support the conclusions?

Reviewer #1: Yes

3. Has the statistical analysis been performed appropriately and rigorously? 

Reviewer #1: Yes

4. Have the authors made all data underlying the findings in their manuscript fully available?

Reviewer #1: Yes

5. Is the manuscript presented in an intelligible fashion and written in standard English?

Reviewer #1: Yes

6. Review Comments to the Author

Reviewer #1: I am satisfied with answers and changes that the authors made. They clarified my questions and provided sufficient rewording to take care of my concerns.

7. PLOS authors have the option to publish the peer review history of their article (what does this mean?). If published, this will include your full peer review and any attached files.

Reviewer #1: No

---

## [Editor Report · Acceptance letter]

26 Jan 2024

PONE-D-23-28449R1 

PLOS ONE

Dear Dr. Gmutza, 

I'm pleased to inform you that your manuscript has been deemed suitable for publication in PLOS ONE. Congratulations! Your manuscript is now being handed over to our production team.

Kind regards, 

on behalf of

Professor Karen Root 

Academic Editor

PLOS ONE